# Characterization of the genome and silk-gland transcriptomes of Darwin's bark spider (*Caerostris darwini*)

Paul L. Babb[1,2], Matjaž Gregorič[3], Nicholas F. Lahens[4], David N. Nicholson[5], Cheryl Y. Hayashi[6], Linden Higgins[7], Matjaž Kuntner[3,8], Ingi Agnarsson[7,9]*, Benjamin F. Voight[1,2,4]*

1 Department of Systems Pharmacology and Translational Therapeutics, Perelman School of Medicine, University of Pennsylvania, Philadelphia, PA, United States of America, 2 Department of Genetics, Perelman School of Medicine, University of Pennsylvania, Philadelphia, PA, United States of America, 3 Jovan Hadži Institute of Biology, Research Centre of the Slovenian Academy of Sciences and Arts, Ljubljana, Slovenia, 4 Institute for Translational Medicine and Therapeutics, Perelman School of Medicine, University of Pennsylvania, Philadelphia, PA, United States of America, 5 Genomics and Computational Biology Graduate Group, University of Pennsylvania, Philadelphia, PA, United States of America, 6 Division of Invertebrate Zoology and Sackler Institute for Comparative Genomics, American Museum of Natural History, New York, NY, United States of America, 7 Department of Biology, University of Vermont, Burlington, VT, United States of America, 8 Department of Organisms and Ecosystems Research, National Institute of Biology, Ljubljana, Slovenia, 9 Faculty of Life and Environmental Sciences, University of Iceland, Reykjavik, Iceland

* iagnarsson@hi.is (IA); bvoight@pennmedicine.upenn.edu (BFV)

**Data Availability Statement:** Data, genome, and transcriptomes generated from this study are available through the central BioProject database at NCBI under project accession PRJNA758099 and

## Abstract

Natural silks crafted by spiders comprise some of the most versatile materials known. Artificial silks–based on the sequences of their natural brethren–replicate some desirable biophysical properties and are increasingly utilized in commercial and medical applications today. To characterize the repertoire of protein sequences giving silks their biophysical properties and to determine the set of expressed genes across each unique silk gland contributing to the formation of natural silks, we report here draft genomic and transcriptomic assemblies of Darwin's bark spider, *Caerostris darwini*, an orb-weaving spider whose dragline is one of the toughest known biomaterials on Earth. We identify at least 31 putative spidroin genes, with expansion of multiple spidroin gene classes relative to the golden orb-weaver, *Trichonephila clavipes*. We observed substantial sharing of spidroin repetitive sequence motifs between species as well as new motifs unique to *C. darwini*. Comparative gene expression analyses across six silk gland isolates in females plus a composite isolate of all silk glands in males demonstrated gland and sex-specific expression of spidroins, facilitating putative assignment of novel spidroin genes to classes. Broad expression of spidroins across silk gland types suggests that silks emanating from a given gland represent composite materials to a greater extent than previously appreciated. We hypothesize that the extraordinary toughness of *C. darwini* major ampullate dragline silk may relate to the unique protein composition of major ampullate spidroins, combined with the relatively high expression of stretchy flagelliform spidroins whose union into a single fiber may be aided by novel motifs and cassettes that act as molecule-binding helices. Our assemblies extend the catalog of sequences and sets of expressed genes that confer the unique biophysical properties observed in natural silks.

is linked to BioSample accessions SAMN21168600- SAMN21168616. All short-read sequencing data has been deposited in the NCBI Sequence Read Archive, and accessions for those data are provided in supplementary materials.

**Funding:** Alfred P. Sloan Foundation (BR2012-087) and NIH Special Instrumentation Grant (1S10OD012312). The funders had no role in the study design, data collection and analysis, decision to publish, or preparation of the manuscript.

**Competing interests:** The authors have declared that no competing interests exist.

## Introduction

Discovered in 2010, the orb-weaving Darwin's bark spider (*Caerostris darwini*) [1] is remarkable not only for the ways it has adapted to use silk for prey capture, but the intrinsic biophysical properties of the silks it weaves to achieve this goal. Produced by the major ampullate silk glands, draglines produced by *C. darwini* can reach the length of 25m and are used to lay bridge lines across waterways to establish capture orbs [2]. To function at this extreme length, draglines produced by *C. darwini* must combine remarkable tensile strength with extensibility to create one of the toughest silks that have been documented to date [3]. In addition, *C. darwini* also creates colossal, prey catching orb webs that are up to $2.8m^2$ in area [1]. The capture spiral of these webs is crafted from material produced in flagelliform silk glands [4], but also requires a viscous adhesive to capture prey *en masse* [1], which is achieved via aqueous secretions produced in the aggregate glands [5–7]. While the collection of these silk properties is defining for all orb-weaving spiders, the extreme nature of the specific properties displayed by *C. darwini* is of fundamental interests across evolutionary, ecological, and material science contexts.

Given the availability of high-throughput sequencing technologies facilitating mass data generation, recent work has greatly expanded the catalog of genomes and transcriptomes available in orb-weaving spider species, including *C. darwini*. We reported the first draft assembly of the genome and transcriptome of the orb-weaving spider, *Trichonephila clavipes* (formerly *Nephila clavipes*). This effort resulted in the first catalog of full-length spider fibroin ('spidroin') genes obtained from the genome with transcription characterized across multiple tissues, including silk glands [8]. Subsequently, genomes for additional orb-weaving spiders were characterized, including for *Araneus ventricosus* [9], the batik golden web spider *Trichonephila antipodiana* [10], the European wasp spider *Argiope bruennichi* [11], and very recently, three additional orb-weavers (*Trichonephila clavata*, *Trichonephila inaurata*, and *Nephila pilipes*) along with an improved draft genome of *T. clavipes* [12]. While this body of work has greatly expanded the repertoire of silk genes available in orb-weavers, the properties and sequences that are specifically utilized by *C. darwini* is not fully understood. Recent long-read transcript sequencing of multiple isolated major ampullate silk glands in *C. darwini* revealed a novel major ampullate spidroin gene (*MaSp4*) highly expressed along with several other spidroin genes [13]. MaSp4 possesses a novel glycine-proline-glycine-proline motif ("GPGP") that may contribute to the toughness of dragline silks [14]. These observations motivate further work to characterize the full genome and transcriptome to elucidate the catalog of spidroins used in *C. darwini* as well as their presumably complex expression across morphologically distinct silk glands.

In what follows, we present draft genomic and transcriptomic assemblies of *C. darwini*, with long-read sequencing follow-up studies to create N and C-Terminal anchored sequences with complete or near complete reconstruction of spidroin/spidroin-like genes. We identified representatives of all major classes of spidroins. Furthermore, we identified expansions of the major ampullate and flagelliform spidroin gene classes, as well as novel spidroin and spidroin-like genes highly expressed in silk glands. Acquisition of tissues from male as well as female specimens also identified sex-specific differences in expression of spidroins, including the elucidation of the subset of spidroins expressed by males with those uniquely expressed in females. We also show substantial expression of flagelliform spidroin genes within major ampullate glands, suggesting that the addition of flagelliform spidroins to major ampullate silk fibers contributes to the exceptional biomechanics of *C. darwini* draglines.

## Materials and methods

### Spider specimens

All specimens from which data were collected derived from wild-caught *C. darwini* adults collected from Andasibe-Mantadia National Park in Madagascar. Details on permit and shipment information are provided (**S1 Table**). DNA for libraries constructed and used for genome assembly was extracted from one female, DNA for long-read SMRT sequencing was extracted from an additional female, RNA for sequencing experiments was extracted from three additional females, and RNA for qPCR validation experiments was extracted from three additional females and three males. The following permits were obtained: Repoblikan'I Madagasikara, Secretariat General, Direction Generale de Forets, Direction de la Conservation de la Biodiversite et due systeme des aires protegees: 090/12/MEF/SG/DGF/DCB.SAP/SCB; Repoblikan'I Madagasikara, Direction Generale des Forets, Direction de la Valorisation des Resources Naturelles, Service de la Gestion Faune et Flore: 042_EA04/MG12; Repoblikan'I Madagasikara, Secretariat General, Direction Generale de Forets, Direction de la Conservation de la Biodiversite et due systeme des aires protegees: 280/17/MEEF/SG/DGF/DSAP/SCB.Re; Repoblikan'I Madagasikara, Secretariat General, Direction Generale des Forets, Direction de la Valorisation des Ressources Forestieres: 314N-EA12/MG17.

### DNA extraction and sequencing

DNA was extracted using phenol:chloroform and column-based methods. Short fragment (180 bp) paired-end sequencing libraries were constructed using TruSeq LT kits (Illumina). Paired-end long-insert jumping libraries were built using two protocols: Illumina's Mate Pair v2 (MPv2) and Nextera Mate Pair kits. MPv2 libraries featured inserts with size ranges of 3 kb, 5 kb, 7 kb, 9 kb and 11 kb, whereas Nextera Mate Pair libraries featured inserts with size ranges of 2 kb, 4 kb, 5 kb, 6 kb, 7 kb, 9 kb, 11 kb, 13 kb, and 17 kb. Libraries were barcoded and pooled for multiple runs per library (**S2**–**S4 Tables**). High-throughput DNA sequencing was performed on either the Illumina HiSeq 2000 or HiSeq2500 (100 x 100) platforms using TruSeq v3 cluster kits and TruSeq SBS chemistry (Illumina).

### RNA-sequencing preparation and data generation

For two individuals, RNA was extracted from the entirety of each specimen for two "whole body" RNA sequencing libraries. For the other individual, select tissues were microdissected (silk glands, venom glands, and brain tissue), and used for 6 tissue-specific RNA sequencing libraries (**S1 and S2 Tables**). In all cases, RNA was extracted using a combined TRIzol (Ambion, Life Technologies) plus column-based protocol. Each of the prepared RNA samples were treated with TURBO-free DNase (Life Technologies), and ribosomal RNA content was depleted with the Ribo-Zero Gold Kit (Epicentre, Human/Mouse/Rat). Strand-specific RNA sequencing libraries were constructed using the NEBnext Ultra Directional RNA Library Prep Kit (NEB, protocol "B") and barcoded using TruSeq RNA adapters (Illumina). All *C. darwini* RNA sequencing libraries are provide (**S2 Table**). High-throughput RNA sequencing was performed on the Illumina HiSeq2000 (100 x 100) platform using TruSeq v3 cluster kits and TruSeq SBS chemistry (Illumina).

### De novo genome assembly

Raw FASTQ read files were evaluated using FastQC (v0.11.268), then trimmed using Trimmomatic (v0.3269) to remove adapter read-through, low quality bases, and ambiguous base calls. All jumping mate pair DNA libraries were processed using the program FastUniq (v1.170) to

remove duplicate read pairs. The *C. darwini* genome was assembled *de novo* using a meta-assembly approach. Three draft assemblies were constructed in parallel using AllPaths-LG vR49967, SOAPdenovo2 (v2.04), and Platanus (v1.2.4), then merged using Metassembler (v1.5) with gaps closed on the metaassembly using GapCloser (v1.12-r6). Genomic quality metrics were calculated for all *C. darwini* assemblies using scripts from the Assemblathon 2 competition [15]. To assess the genome's functional "completeness", the Benchmarking Universal Single-Copy Orthologs (BUSCO) gene mapping method75 was also applied to all *C. darwini* assemblies to identify conserved protein-coding genetic loci [16]. All single copy gene sequences from *Ixodes scapularis* (deer tick) were extracted from the BUSCO Arthropod gene set, a 95% refinement cutoff was applied, and 2,058 *I. scapularis* loci were used to query the completeness of all intermediate and final *C. darwini* assemblies (**S5 Table**).

## De novo transcriptome assembly

After quality control and filtering of reads, all RNA libraries were combined to perform *de novo* assembly as a primary "all-isolate" transcriptome using Trinity (r20140717, **S5 Table**) [17, 18]. Meanwhile, 8 tissue-specific transcriptomes were individually *de novo* assembled using Trinity (**S1, S5, and S6 Tables**). All transcripts were aligned back to the genome using the splice-aware mRNA/EST aligner GMAP (rel_10.22.14) [19], and reads from each RNA library were aligned to the genome using STAR (v2.4.2a, **S6 Table**) [20]. As above, BUSCO was also performed with the all-isolate transcriptome assembly (**S7 Table**).

## Genome annotation

Genomic features were defined on the *C. darwini* final metassembly using four successive rounds of the annotation pipeline MAKER2 [21]. Repetitive regions were identified using RepeatRunner (supplied with Maker2), RepeatMasker v4.0.5 with RMblast, and RepBase repeat libraries [22], then subsequently masked for downstream gene modeling. Tandem repeats were identified using Tandem Repeats Finder v4.07b [23]. Gene models were based on multiple types of evidence: alternate species protein sequence alignments, alternate species EST/mRNA/cDNA sequence alignments, *de novo* assembled transcripts from *C. darwini* RNA-seq experiments, and *ab initio* gene predictions. Protein and EST/mRNA sequences were collected from online databases (**S8 Table**). Exon boundaries were marked using Exonerate v2.2.0 [24], and tRNA by tRNAscan-SE [25]. Feature boundaries were further polished by Maker2 directing successive rounds of trained predictions from SNAP (rel_11.29.13) [26] and Augustus v3.0.2 (WebAugustus41) [27]. In total, >9 million genomic features and 410,081 putative genes were modeled on the *C. darwini* final annotated meta-assembly (**S9 Table**). Putative gene model identities were established by the reciprocal alignment of model protein sequences to the UniProtKB/Swiss-Prot85 protein database (v6.3.15) using BLASTP86 and Maker2 accessory scripts. Five tiers/sets of gene models with increasing stringency were defined on the basis of agreement among coding feature annotations, conserved protein domains, eukaryotic gene structure, and similarities with curated gene databases (**S9 Table**). Downstream analyses were based on the "Gold" gene set (14,894 genes, 15,343 mRNAs), and contained only gene models that possessed known protein domains from the InterPro Pfam database [28], was produced using BLASTP [29], HMMer [30], and "Maker Standard" scripts (K. Childs), but also had biological supporting evidence (RNA, protein alignment).

## Spidroin identification and validation

*C. darwini* spidroins were identified by multiple BLAST [29] searches of the genome, transcriptomes, and gene models using a previously curated database of spidroin query sequences

(S8 and S10 Tables). We assessed sequence homology to known spidroin sequences by aligning Caerostris darwini genomic scaffold sequences using different BLAST algorithms (BLAST v2.2.29: blastn, tblastx, tblastn) in multiple successive rounds of the different genomic sub-assemblies and the final meta-assembly. Query sequences (524 nucleotide and 380 amino acid sequences) were collected from GenBank, literature review, and the Trichonephila clavipes spidroin catalog, and were compared using a BLAST e-value threshold of 1E-6. Positive alignments were tallied, and the scaffolds and gene models with manually inspected for additional spidroin signatures (the "MAFAS" motif in N-terminal domain, repetitive GPGQQ or poly-A coding motifs, and evidence of expression from RNA-seq reads in silk gene tissues). Caerostris darwini sequences that had support from multiple lines of evidence (homology, known motifs, gene expression profiles) were promoted as candidate spidroin loci and subjected to further analyses such as motif painting, phylogenetic comparisons, qPCR gene expression profiling.

Five loci exhibited complete coding sequences, but the remainder of putative *C. darwini* spidroins had internal sequence gaps, were only repeats, or were incomplete N- or C-terminal sequences at the end of scaffolds. To identify missing pieces, multiple rounds of searching was performed by adding *C. darwini* spidroin hits from the previous round to each new list of queries. Putative spidroin fragments were organized into five categories for validation and completion experiments (see below): complete, internal gap, 5′ end, 3′ end, and repetitive sequence (S11 and S12 Tables).

Putative *C. darwini* spidroins were isolated and filled using a combination of Long-Range PCR (LR-PCR) followed by Single Molecule Real-Time (SMRT) sequencing of a single *C. darwini* adult female (Cae-009, S2 Table) at very high coverage. Multiple pairs of LR-PCR primers (S11 Table) were designed for each scaffold (Primer3), so that putative spidroin loci could be completely isolated by LR-PCR amplicons, and alternate primer pairs could be recruited in cases of sub-optimal amplification. Pair-mates were proposed using sequence similarity, orthologous alignments, and transcript tissue specificity. To "bridge" two separate scaffolds, multiple combinations of cross-pair LR-PCR experiments were performed to identify scaffold pairs that were more cryptically related. LR-PCR reactions employed high efficiency PrimeStar GXL polymerase (Clontech/TaKaRa) and visualized on low voltage 0.5% Bio-Rad Certified Megabase agarose gels. Amplicons were purified and pooled at equimolar ratios, with slightly higher volumes for the longest fragments (>20 kb). Two unique pools of spidroin amplicons were processed for SMRT library construction [31], and sequenced using the P6-C4 sequencing enzyme, chemistry, and 4-hour movie collection parameters (Pacific Biosciences). Quality-filtered FASTQ files of long SMRT reads were directly aligned to scaffolds that exhibited complete spidroins or spidroins with internal gaps on single scaffolds using PBJelly (PBSuite 15.2.20.p1) [32] and BLASR (v1.3.191) [33]. For putatively linked scaffold pairs, manual alignments were performed to effectively bridge gaps, correct errors, and resolve repeats. In total, 23 *C. darwini* spidroin sequences and 3 spidroin-like sequences were validated (Fig 1 and S12 Table).

## Spidroin gene repeat motif identification and analysis

An initial set of *C. darwini* spidroins were translated into amino acid residues (S1 Data) and then subjected to repeat motif identification using MEME and motif painting with MAST v4.10 [34]. Repetitive motifs were manually curated to remove low-quality hits and motifs occurring in N- and C-terminal domains (using a hard cutoff of 100 residues) and cataloged as unique "motif variants" ranging from 4 to 41 residues in length. Motif variants were then organized into "motif groups" based on residue content and sequence (S13 Table). Motifs that could not be informatively grouped were designated "unassigned". The full catalog of motif

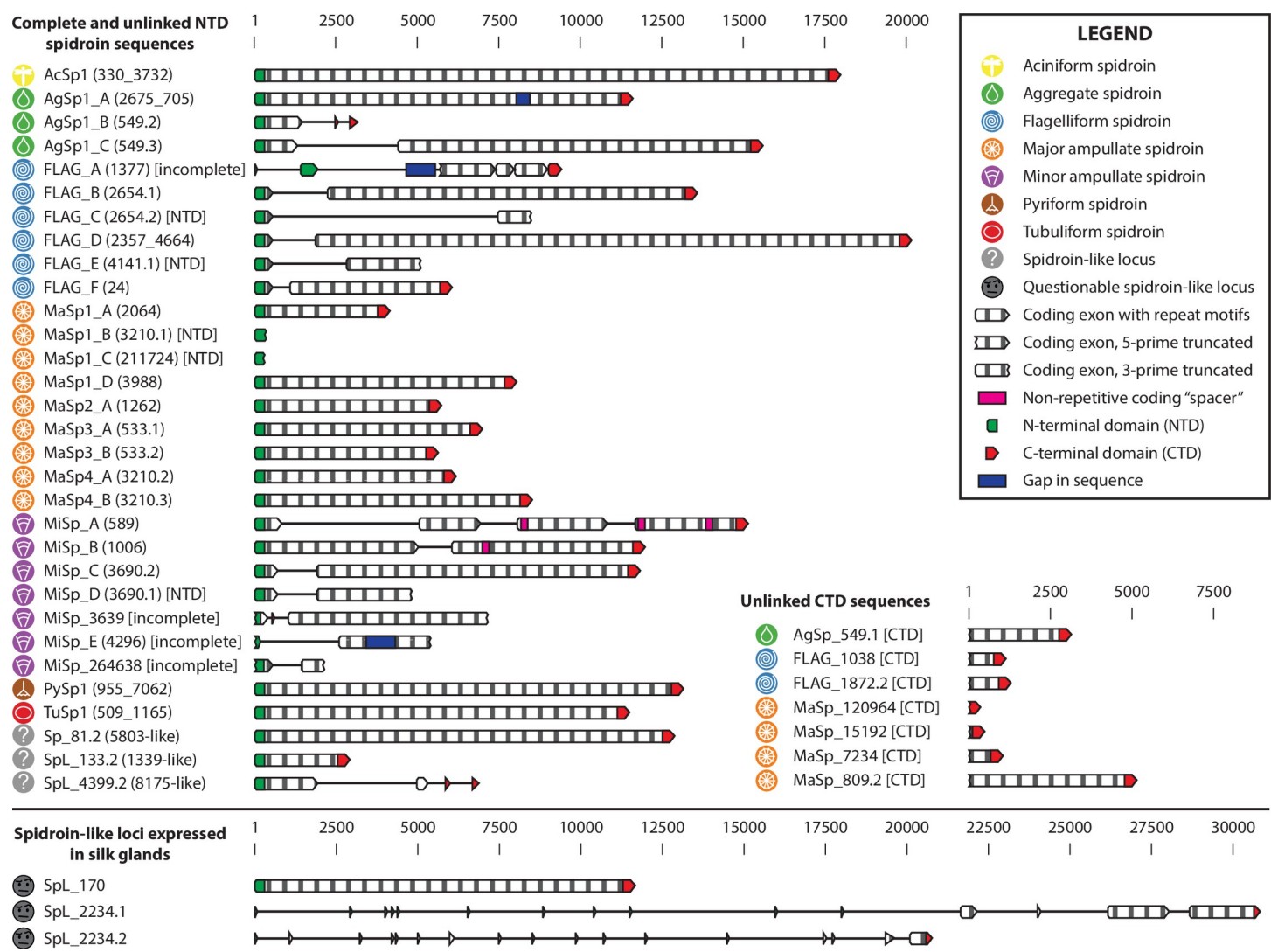

**Fig 1. A catalog of spidroin genes identified in *Caerostris darwini*.** Circular symbols denote the silk class of each spidroin, determined by alignment of N and C-terminal domains and motifs to previously known spidroin sequences. Genic structures are drawn to scale. Scaffold identifiers for the locations of the genes are provided in parentheticals, or embedded in the name (for Spidroin-like sequences). AcSp: Aciniform, AgSp: aggregate; FLAG: flagelliform; MaSp: major ampullate; MiSp: minor ampullate; PySp: pyriform; TuSp: Tubuliform; SpL: Spidroin-like.

variants was input in secondary rounds of motif searching with the custom pipeline (see **URLs**; **S13 Table**). Next, the pipeline was used to search for higher-order repetitive structures denoted as ensembles / cassettes, defined at least two adjacent motif occurrences that were enriched across spidroins. Cassette variants were organized into "cassette groups" and curated to remove cassette types that exhibited inter-motif gaps >20 residues or that occurred in N- and C-terminal domains (**S14–S19 Tables**). Due to a technical oversight, *SpL_170* was not included in this motif and cassette discovery analysis.

## 2D structure analysis

Using protein translated sequences for our assembled *MaSp4_A* and *MaSp4_B* transcripts, we utilized the RaptorX-Property web server [35] to generate 8-state 2-dimensional secondary protein structure predictions, with default parameters utilized.

## Quantitative PCR (qPCR) analysis

For the specimens used for these experiments (see above), from the abdomen, silk glands were identified by relative position and morphology, then individually collected by severing their ducts near the spinnerets. From the cephalothorax, legs were collected, venom glands were collected after separation of the chelicerae from the cephalothorax, and the remaining cephalothorax tissue was retained as the "head" sample. In total, each specimen was microdissected into nine tissue subsections: venom glands, head (with no venom glands), legs, major ampullate silk gland (MA), minor ampullate silk gland (MI), flagelliform silk gland (FL), aggregate silk gland (AG), tubuliform silk glands, and "other silk glands" (OTHER: piriform and aciniform glands, attached to spinneret) which yielded 27 experimental samples in total (**S1 Table**). RNA was extracted using TRIzol (Ambion, Life Technologies) and RNeasy Mini Kit spin columns (Qiagen), and additional cleanup performed using the RNA Clean & Concentrator-5 kit with DNAse I treatment (Zymo Research). Small aliquots (~5 μL) were used for quality control and quantification. cDNA was produced from each RNA sample with a high capacity cDNA Reverse Transcription kit (Life Technologies) and run alongside multiple "no reverse transcriptase" [NRT] negative controls. Primers were designed to target 31 loci (all 23 spidroins, six spidroin-like genes, one venom locus [CRiSP/Allergen/PR-1] [36], and one housekeeping gene [RPL13a]), plus genomic-scaffold-controls for all single-exon spidroin genes (**S20 Table**). qPCR reactions were set up in triplicate using standard SYBR Green PCR Master Mix (Life Technologies) and run on a ViiA 7 Real-Time PCR machine. Relative transcript abundance of targets in silk and venom gland samples was normalized to leg tissue samples and calculated using the $2^{-\Delta\Delta CT}$ method [37].

# Results

## Draft, annotated genomic and transcriptomic assemblies for *Caerostris darwini*

We began by sequencing genomic DNA isolated from a field-collected *C. darwini* female generated from a diverse set of sequencing libraries we prepared (**Methods, Spider specimens and DNA extraction and sequencing**). Next, we analyzed the data generated from these libraries using multiple computational *de novo* genome construction methods, unified the results via meta-assembly, and constructed 1.45 Gb of draft genome (**Tables 1 and S1–S4**). We estimated the size of the full *C. darwini* genome to be 1.81 Gb. Our meta-assembly consisted of 45,784 scaffolds (N50: 489.8 kb, N50 contig size: 94.6 kb), with ~140x coverage from re-mapping of 1.34 billion 100 bp quality control, filtered, unique read-pairs (**S5 Table**).

To map the locations of protein coding and transcribed genes within the *C. darwini* genome, we extracted RNA from multiple isolated tissues–whole body, brain, venom and silk glands–that were obtained from three field-collected females (**Methods, Spider specimens**). We obtained strand-specific 100-bp paired-end reads from RNA-sequencing and performed computational *de novo* assembly for each isolate (**Methods, *De novo transcriptome assembly***). We also assembled a transcriptome representing the union of RNA-Seq data from all the isolated tissues, which comprised ~309 million quality control, filtered unique read-pairs (**Tables 1 and S6 and S7**). To quantify the completeness of the protein-coding genome, we searched our draft assemblies for homology to 1,066 curated arthropod sequences [8], and estimated that our draft genome is 95.2% complete and our all-isolate transcriptome assembly is 97.5% complete (**S5 and S7 Tables**).

Finally, we applied the MAKER2 pipeline [21] to enrich our draft genome with annotations. Our annotation mapping efforts were informed by coding sequences from a catalog of data

**Table 1. Summary statistics for the *C. darwini* genome and transcriptome assemblies.**

| Estimated Genome Size | | |
|---|---|---|
| Genome Size[a]: | 1.81 Gb | |
| **Genome Assembly** | **"Full"[b]** | **"Annotated"[c]** |
| Assembly Size: | 1.53 Gb | 1.45 Gb |
| | 1.51 Gb non-gap | 1.43 Gb non-gap |
| % Genome Captured: | 84.9% | 80.4% |
| Number of Contigs: | 253,859 | 465,207 |
| N50 Contig Size: | 87,597 bp | 94,553 bp |
| Number of Scaffolds: | 232,896 | 45,784 |
| N50 Scaffold Size: | 453,395 bp | 489,784 bp |
| Largest Scaffold: | 4,645,134 bp | 4,645,134 bp |
| Scaffolds >100 kb: | 3082 | 3082 |
| BUSCO % recovered[e]: | 94.8% | 95.2% |
| **Transcriptome Assembly** | **"All Isolates"** | |
| Read Input: | $6.19 \times 10^8$ reads | |
| Number of Transcripts: | 1,056,281 | |
| N50 Transcript Contig Size: | 911 bp | |
| BUSCO % recovered[e]: | 97.5% | |

Statistics regarding construction of the draft meta-assembled genome and "all isolates" transcriptome:

[a] genome size estimate calculated based on k-mer frequency (K = 25 scale);

[b] gap-closed meta-assembly of AllPaths LG + SOAPdenovo2 + Platanus (minimum scaffold length = 100 bp);

[c] gap-closed meta-assembly of AllPaths LG + SOAPdenovo2 + Platanus (minimum scaffold length = 1,000 bp + 49 additional scaffolds containing BLAST hits for previously published spider spidroin gene sequences);

[d] unique QC-filtered paired and single reads remapped to assembly;

[e] completeness based upon matches to 2,058 *I. scapularis* BUSCO loci. Additional genome assembly metrics are provided in **S5** and **S7 Tables** for transcriptome metrics.

from closely related species (**S8 Table**), the all-isolate transcriptomes that we generated, application of *in silico* gene prediction methods, and libraries of transposable elements and repeated motifs (**Methods, Genome annotation**). In total, we attached >12 million features to our draft assembly, and conservatively estimated 14,894 genes present in the genome ("Gold" annotation, **S9 Table**).

## Identification of the *C. darwini* spidroin gene catalog

Anticipating that our draft assembly would not fully capture the complete sequences for spidroins–a class of very large, highly coding-repetitive genes–we next performed validation of our assemblies as well as locus refinement for spidroin genes. To achieve this goal, we first searched our draft genomic and transcriptomic assemblies as well as our annotated gene models for sequences that aligned with a curated library of previously established spidroin genes (**Methods, *Spidroin identification and validation*, S10 Table**). After filtering (**Methods**), we identified 34 candidate N-terminal domain (NTD) and 33 candidate C-terminal domain (CTD) sequences from this search. Using this set of candidate sequences, we next performed long-range PCR amplification followed by single-molecule real-time sequencing (SMRT Pac-Bio) to reconstruct full length sequences at high coverage (**Methods, S11 Table**). These data allowed us to link NTD and CTD on a single scaffold for 23 spidroins and 3 spidroin-like genes, based on definitions described previously [38], with partial sequences for the remaining 8 NTD and 7 CTD sequences (**Fig 1**). While gaps persisted in this subset, we were able to

enumerate substantial portions of their repeated motif structures, which we characterize in more detail below.

To map our spidroins to the set of the major silk gene classes that have been previously described, we performed sequence alignment of the conserved NTD and CTD as well as comparing to internal motifs reported to be specific to a particular spidroin class [8]. To aid in clarifying cryptic, unclassified spidroin-like sequences, we used the *T. clavipes* spidroin sequence catalog we previously generated. We observed that the majority of spidroins identified in *C. darwini* map to one of the seven previously categorized spidroin classes (**Fig 1**). Within these classes, we identified novel members, which expanded several classes in *C. darwini*. Relative to *T. clavipes*, we found one additional major ampullate spidroin (MaSp), three additional minor ampullate spidroins (MiSp), and four additional flagelliform spidroins (FLAG). The expansion of FLAG genes was intriguing, as this class of genes is used by spiders in silks to construct the orb web's "stretchy" capture spiral, suggesting perhaps additional adaptive pressure on these classes of genes in this lineage as it diverged from the common ancestor with *T. clavipes* [39–41]. However, three *C. darwini* spidroins–Sp_81.2, SpL_133.2, and SpL_4399.2 –eluded assignment to any of the previously established classes based on sequence homology alone.

## A catalog of spidroin repetitive motifs and cassettes in *C. darwini*

Using the catalog of spidroins identified in *C. darwini* and *T. clavipes*, we sought to characterize the set of coding motifs found within these genes that were unique to or shared between these species. To achieve this, we applied the computational motif detection and demarcation approach we previously described [8], working first to enumerate the complete set of repetitive sequence motifs with catalogs of spidroin and spidroin-like genes described above along with those we previously reported in *T. clavipes* (**Methods, *Spidroin gene repeat motif identification and analysis***). Overall, we observed 11,024 occurrences of 2,771 motif sequences found in both species, ranging from 4 to 41 amino acids in length (**Tables 2 and S12 and S13**). 6,950 (63%) of occurrences were found in *C. darwini*, likely due to the larger repertoire of more complete spidroins characterized from *C. darwini* than *T. clavipes*. 1,493 (53%) of motif variants enumerated were exclusive to *C. darwini* (**Table 2**), with the majority of those (1189 of 1493, 79.6%) originating from a specific *C. darwini* gene with a proportion similar to *T. clavipes* (301 of 403, 74.6% **S14 and S15 Tables**).

The subset of the 2,711 motif types that were specific to *C. darwini* but shared across genes within class (n = 127, 8.5%) or across classes (n = 177, 11.9) were the most frequent, indicating motif types that are heavily utilized by *C. darwini*. Of those, the greatest number were observed in MaSp class members which included several of the xPGPQ motif varieties found specifically

**Table 2. Spidroin repeat motif summary for *C. darwini* and motif sharing with *T. clavipes*.**

| Metric | Count |
| --- | --- |
| Number of Motif Variant Sequences types detected | 2771 |
| Types private to *C. darwini* (%) | 1493 (53.9) |
| Types private to *T. clavipes* (%) | 403 (14.5) |
| Shared between *C. darwini* and *T. clavipes* (%) | 875 (31.6) |
| Number of defined Motif Groups | 140 |
| Number of defined Motif Sub-groups | 302 |
| Number after variant descriptions collapsed | 246 |
| Total Motif Occurrences observed for n = 38 *C. darwini* spidroin or spidroin-like genes | 6950 |
| Total Motif Occurrences observed for n = 28 *T. clavipes* spidroin or spidroin-like genes | 4074 |

in both MaSp4 spidroins (*MaSp4_A* and *MaSp4_B*) and the CTD of another putative member of the MaSp class (**Fig 2**). Both MaSp4 genes were described by numerous tandem repeats of GPGPQGPS, flanked by Serine, Valine, and Threonine-rich motifs (e.g., SVSVVSTTIS VSVVSTTVS) vaguely analogous to homopolymer runs of alanine common to minor ampullate spidroins. 2D protein structural analysis (**Methods, *2D structure analysis***) suggests these spacers may form beta-strands that can, when adjacent to one another, form hydrogen bonds leading to formation of beta-sheets. In addition, the FLAG class members in *C. darwini* had many repetitive motifs that were shared across this class specifically. The most common *C. darwini* specific motifs found were [GGP]₃ and GGSGGGL, repetitive sequences not enumerated in *T. clavipes* (**Fig 2**). These motifs have been previously hypothesized to link together to form structures that might also form intramolecular bonds contributing to elasticity [42].

| Motif Group Name | Motif | Number of Occurrences (*C. darwini* / *T. clavipes*) | Found In Spidroin Classes *C. darwini* | *T. clavipes* |
|---|---|---|---|---|
| **Private to *C. darwini*** | | | | |
| xPGPQ containing | GPSGPGPQ | 110 | Major ampullate | - |
| | [GPGPQ]₂ | 72 | Major ampullate | - |
| | GPYGPGG | 64 | Major ampullate | - |
| | SPGPQ | 49 | Major ampullate | - |
| | GPYGPGPQ | 47 | Major ampullate | - |
| | GPSGPGSQ | 33 | Major ampullate | - |
| Multiple xGG | [GGP]₃ | 37 | Flagelliform | - |
| | GGYPGGPDY | 28 | Aciniform | - |
| GL, GGL, GGGL containing | GGSGGGL | 30 | Major ampullate, Flagelliform | - |
| GTGGTTGT containing | GTGGTTGTGDSDTP | 34 | Questionable Spidroin-like | - |
| L+S Rich | SELLVSLAGIS | 31 | Aciniform, Aggregate | - |
| GGY, GGGY containing | GGYTGIPS | 30 | Aciniform | - |
| N+Q+S Rich | QSDKNLV | 30 | Aciniform | - |
| PxSDARVxxxVxVSACVxAMLSSG containing | PNSDARVCANVIVSACVKAMLSSG | 29 | Aciniform | - |
| DAVCGAAxRAGxyIPDSVV containing | DAVCGAAGRAGMRIPDSVV | 27 | Aciniform | - |
| **Shared with *T. clavipes*** | | | | |
| Multiple xGG | GGYGPGGQ | 101 (14 / 87) | Major ampullate | Major ampullate |
| Stretches of Alanine with GA repetitive flanks | AAAAA[GA]₃GGA | 77 (68 / 9) | Minor ampullate | Major ampullate, Minor ampullate |
| | [AG]₃AAAAA[GA]₂GGA | 25 (18 / 7) | Minor ampullate | Minor ampullate |
| MTTS containing | MTTSATTPA | 60 (21 / 39) | Major ampullate, Spidroin-like (81.2) | Major ampullate, Spidroin-like (5803) |
| GPGxQ containing | GPGSQGP | 48 (9 / 39) | Major ampullate, Aggregate | Major ampullate |
| | GPGPQGPSGP | 33 (7 / 26) | Major ampullate, Aggregate | Major ampullate, Aggregate |
| GGY, GGGY containing | GGYGAGSGG | 37 (34 / 3) | Major ampullate, Minor ampullate (133.2) | Major ampullate, Minor ampullate |
| | GGYGGG | 24 (20 / 4) | Minor ampullate, Spidroin-like | Minor ampullate |
| GQGGY containing | GYGGQGGY | 25 (6 / 19) | Major ampullate, Flagelliform | Flagelliform, Minor ampullate |
| Stretches of Alanine, with substitutions | AAAAAAAT | 33 (23 / 10) | Major ampullate, Minor ampullate | Major ampullate |
| | SAAAAAAA | 31 (2 / 29) | Major ampullate | Major ampullate |
| Multiple instances of GA repeat | AGS[GA]₅GGA | 33 (27 / 6) | Minor ampullate, Spidroin-like (81.2) | Minor ampullate, Spidroin-like (5803) |
| GTPxxTSSGGSx containing | GTPASTSSGGSP | 32 (9 /23) | Spidroin-like (81.2) | Spidroin-like (5803) |
| Serine Rich | STPSQ | 27 (20 /7) | Major ampullate, Spidroin-like (81.2) | Pyriform, Spidroin-like |
| Multiple xGG | GGQ[GGP]₂ | 27 (22/ 5) | Flagelliform | Major ampullate |

KEY: Aciniform · Aggregate · Flagelliform · Major ampullate · Minor ampullate · Pyriform · Spidroin-like · Questionable Spidroin-like

**Fig 2. Most frequent coding repeat sequence motifs found only in *Caerostris darwini* or shared with *Trichonephila clavipes*.** Top 15 most frequently observed motifs that were exclusive to *C. darwini* (upper frame) or shared with *T. clavipes* (lower frame). For motifs found in spidroin-like genes or atypical spidroin genes, a brief label for the name of the gene is provided (e.g., 81.2, 133.2, and 5803). The single example where motif usage was shared but utilized in different spidroin classes is shaded.

Beyond the substantial number of unique motifs that were Glycine, Serine, and Proline-rich, a handful of other unique motifs were also present, which included SGMMY, TVVDN[L|I] SVNV, and PITISGTLNV (S13 Table). Finally, the aciniform spidroins appeared unusual in the distinctive divergence of repetitive motif sequences utilized across both species. In *C. darwini*, AcSp1 was a diversely repetitive spidroin containing over 300 motif occurrences of 16 motif types (into 11 distinctive structures, S14 Table) including several that were quite long (7–24 amino acids in length, Fig 2). However, none of these motif types were observed in *T. clavipes*, which contained 162 motif occurrences across 14 distinct types, only two motifs of which were shared across aciniform spidroins between species (n = 15 occurrences total, S14 Table). This divergence, previously noted in other species for aciniform silk [43], is consistent with strong, adaptive evolution on this spidroin.

The motifs shared between *C. darwini* and *T. clavipes* tended to be present in similar classes of spidroin genes in both species (Fig 2). For example, as expected given previously described spidroins across spider species, the most frequently shared motifs across these species included the previously described catalog of Glycine-rich motifs (e.g., GYGGQ, GGY, etc.) for flagelliform spidroins, stretches of poly-alanine repeats in major and minor ampullate spidroins, and GA repeats in minor ampullate spidroins (Fig 2). In addition, we noted similarities in the frequency of use for shared motifs between two spidroins (Sp_5803 in *T. clavipes*, Sp_81.2 in *C. darwini*). Sp_5803 is highly expressed in flagelliform glands (as is Sp_81.2 in *C. darwini*, see below); thus, the frequently shared use in these spidroins suggests conservation of elements to preserve its function. In contrast to use of motifs across species with spidroin classes, we noted a commonly occurring motif (GGQ[GGP]$_2$) which appears to be utilized in major ampullate spidroins in *T. clavipes* but flagelliform in *C. darwini*. We note that this repeat was quite like the commonly occurring motif [GGP]$_3$ described above, which was also found in flagelliform spidroins in *C. darwini*. Given their suggested 2D protein structural properties [42], these structures and motifs may further contribute to the extensibility of this silk.

We next used the catalog of short motifs described above to enumerate the set of coding sequences comprised of multiple motifs in tandem (referred to as ensembles [44, 45] or cassettes [8]) found in *C. darwini* and *T. clavipes* spidroin genes. To achieve this, we utilized our previously described computational cassette detection and demarcation approach [8], working to apply the lists of repetitive sequence motifs jointly to the set of spidroin and spidroin-like genes identified in both spider species (**Methods, *Spidroin gene repeat motif identification and analysis***). Overall, we observed 3,693 occurrences across 2,268 cassette sequence types found in both species (S16 **and** S17 **Tables**) comprised of 2 to 6 motifs each (S18 Table). 2,344 (63.4%) of cassette occurrences were found in *C. darwini*, again presumably due to the larger catalog of spidroin genes overall identified. We observed only a small fraction of cassettes shared between *C. darwini* and *T. clavipes* (67 / 3,693 = 1.8%, S19 Table), which contrasted with the much greater proportion of non-tandem motifs shared between them (= 31.4%, Table 2). Not surprisingly, the most frequent cassettes across classes were those motifs that were common to the class, e.g., the DTxSYzTGEY motif, common to aggregate spidroins, was also a common component of cassettes found in aggregate spidroins; similarly, cassettes containing poly-Alanine tracks were abundant in minor ampullate spidroins (S17 Table). Flagelliform spidroins had by far the most diverse number of cassette types (n = 10) involving 308 occurrences of the xGG_GGx cassette, a cassette used across virtually all spidroin gene classes. Consistent with the presence of motif frequency seen above, we also found GPGPQ in tandem (n = 57) and poly-Alanine+GPGPQ cassettes (n = 19) frequently in *C. darwini* and present in MaSp4 and AgSp1. Taken collectively, these data suggest that the repertoire of repetitive motifs combine uniquely and amplify in number in a species-specific way.

## Tissue-specific expression of each spidroin across silk glands

Previous work has demonstrated that spidroins assigned to a particular gene class are not exclusively transcribed in the silk gland in which the gene is assigned [8, 13] with proteomic analyses also indicating diverse protein expression across silk glands [46]. To validate and compare transcription levels across silk glands, we directly interrogated the degree of spidroin expression bias by using quantitative PCR (qPCR) to measure RNA transcript levels in morphologically classified silk gland isolates and control tissues isolated from three field-collected adult females (**Methods, *Quantitative PCR (qPCR) analysis***). We were able to cleanly separate isolates of all morphologically distinct gland types, except for the aciniform and piriform glands, which due to their proximal anatomic locations (i.e., attached to the spinnerets) and small size could not be cleanly separated and were therefore treated as a combined sample ("Other Silk Glands", **Methods, *Quantitative PCR (qPCR) analysis***). In total, we profiled 25 spidroin genes, 3 spidroin like genes with canonical N- or C-terminal domains, 3 additional spidroin-like genes, and 2 control genes (*PR-1*, for venom gland expression and *RPL13a* as a normalizing control) with 2 technical replicates per tissue isolate (**Methods, *Quantitative PCR (qPCR) analysis***).

Profiling expression across tissues revealed both ubiquitously and gland-specific spidroin expression in *C. darwini*. Consistent with patterns of expression previously characterized in other spider species, we observed examples of spidroin genes from each class that are highly expressed in their corresponding morphologically distinct silk gland. This included, but was not limited to, *MaSp1_B* in major ampullate glands (**Fig 3A**) or *Flag_D* in flagelliform glands (**Fig 3B**). In addition, we observed that in each silk gland assayed, spidroin transcripts belonging to more than one silk class were also detected (**Figs 3A and 3B and S1 and S2**) but with very low expression in non-silk gland tissues (**S3–S5 Figs**). Among those spidroins that were highly expressed across a range of silk glands were *AgSp1_B*, *MaSp1_B*, *MaSp4_A*, and to a lesser extent, *MaSp4_B* (**S1 and S2 Figs**). Despite observing relatively broad expression for most spidroin genes, we noted two transcripts whose expression was primarily found in a single gland, and not strongly expressed in others: *MaSp1_A*, strongly expressed in major ampullate, and *Flag_F*, which was modestly expressed above baseline in flagelliform glands only. In addition, we also noted two transcripts whose highest expression did not match the N- and C-Terminal similarity class definition: *MaSp2_A*, whose expression was highest in aciniform/pyriform glands, and *MiSp_E*, whose expression was highest in tubuliform and flagelliform glands. These data continue to support a model whereby natural silk extruded by orb-weaving spiders for specific needs is the composite of many spidroins expressed across many glands.

We next aimed to use transcriptional expression as information to guide the classification of the spidroin-like transcripts we had identified and assembled. We found that the glands in which these spidroin-like transcripts had the highest expression were: aciniform/pyriform glands for *SpL_133.2* and aggregate glands for *SpL_4399.2*. For the remaining non-canonical spidroin-like genes, those also include flagelliform glands for *SpL_2234.1* and *SpL_2234.2*, and major ampullate (by a very small amount over minor ampullate) gland for *SpL_170*. We also note that the atypical spidroin, Sp_81.2, was most highly expressed in flagelliform glands. These patterns of expression perhaps give a small clue to the nature and possible role of these genes in natural silk production, either directly in silk fibers or in genes to facilitate assembly or aggregation in the gland.

The gene expression profiling results present here thus far derived from mature females. To elucidate the repertoire of spidroin genes utilized in males versus females, we next performed qPCR to profile expression of the above collection of spidroin and spidroin-like transcripts in silk gland and control tissues isolated from field-collected mature males (**Methods,**

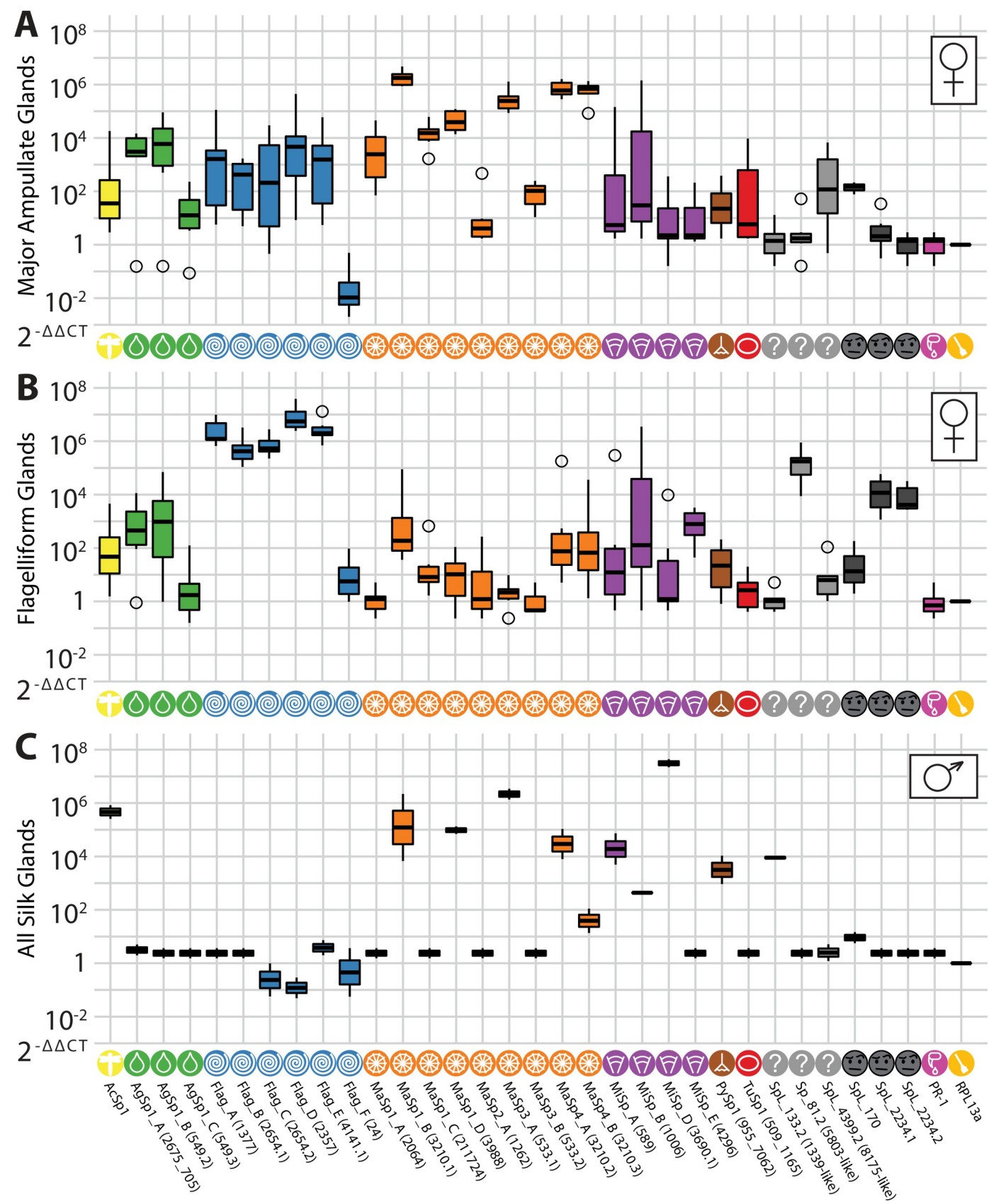

**Fig 3. Results of qPCR Expression profiling of all spidroins across silk glands in female and male specimens.** Relative transcript abundance of spidroin targets plus controls across silk gland samples, normalized to leg tissue samples and calculated using the 2-ΔΔCT method (**Methods**). Results reported are specifically for (**A**) major ampullate and (**B**) flagelliform glands in females, as well as (**C**) all silk glands in males.

***Quantitative PCR (qPCR) analysis***). Given the extreme sexual size dimorphism in *C. darwini* with males >3 times smaller than females [1], it was not feasible to dissect individual silk glands from males; thus, we profiled abdomens and a combined sample of extracted silk glands instead (**Figs 3C and S6**) in addition to non-silk gland tissue isolates (**S7 and S8 Figs**). First, we observed little if any expression of aggregate, flagelliform, or tubuliform spidroin transcripts. This was not altogether unexpected given the contribution of these classes of genes to construction of the web capture spiral (flagelliform), prey capture using the capture spiral (aggregate), or egg-case construction (tubuliform), which are behaviors not utilized by mature males [38]. In fact, male *C. darwini*—as do male spiders in general—lack tubuliform spigots or associated glands and do not appear to have a functional 'triad' of a flagelliform and two aggregate spigots (**S9 Fig**). This reflects the biological reality of sexual dimorphism in most orb web spiders, where adult males do not engage in web building but rather inhabit female webs awaiting mating encounters and feeding along with the females [47]. Our own field observations confirm this to be true for *C. darwini* [1]. In addition, we noted that only a subset of major or minor ampullate spidroins were expressed (**Fig 3C**), suggesting utilization of these transcripts in males for their limited silk production capabilities, with the complement of those expressed in females (*MaSp1_A*, *MaSp1_C*, *MaSp2_A*, *MaSp3_B*, and *MiSp_E*) suggestive of the importance of these transcripts for the substantially more diverse repertoire of silks constructed by females. As above, this finding reflects the species biology where males do not construct capture webs. Among the set of spidroin-like transcripts, we noted high expression of *SpL_133.2*, further supporting its novelty and additional co-expression with *AcSp1* and *PySp1* perhaps suggestive of its functional role.

Curiously, we noted modest, but non-trivial expression of several spidroins outside of silk glands in males. First, *MiSp_D* appeared to have non-trivial expression in the cephalothorax, chelicerae plus venom gland, and pedipalps, correlating with the expression of the venom-gland control gene *PR-1*. Similarly, a handful of flagelliform spidroins (*Flag_A*, *Flag_B*, *Flag_E*) also appeared slightly upregulated in cephalothorax and chelicerae plus venom gland. Finally, *MiSp_E* appeared to be expressed in male pedipalps specifically. Given previous examples of canonical spidroins transcribed outside of silk glands in other species [8], these patterns of expression hint at the possibility of repurposing of spidroin genes.

## Discussion

In summary, we created an initial draft, annotated genomic and multi-tissue transcriptomic assemblies for *C. darwini*. These assemblies enabled the identification of 31 putative spidroin genes, including several novel transcripts. Validation and gap filling using qPCR followed by long-read sequencing allowed us to characterize the patterns of spidroin motifs found in *C. darwini* and by comparing them to those of *T. clavipes* enabled the identification of candidate motifs that contribute to the unique properties of silks woven by *C. darwini*. Our gene expression profiling demonstrated examples of gland-specific expression of spidroin genes, as well as spidroins broadly expressed across multiple glands. On this particular point, we note very recent work which reached similar conclusions from analysis of protein expression of silk glands in orb-weaving spiders as well [46], further supporting that diverse spidroin expression is functionally consequential. Finally, characterization of expression in males elucidated the

catalog of spidroins utilized by males, and the complement of spidroins utilized by females, including multiple *MaSp* and *Flag* transcripts.

Recent work has greatly expanded the number of spider genomes available for analysis, and thus considering how *C. darwini* relates to *T. clavipes* bears some brief discussion. Currently, *C. darwini* and *T. clavipes* are catalogued as representatives of the same family, *Araneidae* [48]. However, Kuntner et al. [49] consider *Trichonephila* as member of *Nephilidae*, and *Caerostris* as member of *Araneidae*, both families being true orb-weavers, *Orbipurae*. This is because Araneidae as currently circumscribed is poorly defined and ancient [49]. Regardless of which family classification is being followed, the current evidence dates the most recent common ancestor of *Caerostris darwini* and *Trichonephila clavipes* to between 62 and >200 million years ago, albeit with large confidence intervals [40, 49].

The function and nature of Sp_5803 as a spidroin itself has not been clear, and our results here provide further support for this target as a true spidroin. Detailed examination of the domains of Sp_5803 in *T. clavipes* with comparative genomics across available spider species revealed that this spidroin has only the canonical N-terminal domain region mapping to previously characterized true spidroins, but does contain a high density of ordered repeats typical of spidroins. Sp_5803 was highly expressed in the flagelliform glands of *T. clavipes* females but not males [8, 38]. In *C. darwini*, we observed a gene (*SpL_81.2*) with features quite similar to *Sp_5803* in that it was also missing a canonical spidroin C-terminal domain while carrying a canonical spidroin N-terminal domain, sharing canonical sequence motifs with *Sp_5803* that feature dense sets of repeats typical of a spidroin (**Fig 2**), and is highly expressed in the flagelliform glands of females but not males (**Fig 3B**). We thus propose that *T. clavipes Sp_*5803 and *C. darwini Sp_81*.2 are orthologous spidroin genes. Further comparative genomics analysis with newly available orb-weaving spiders could provide additional data on this hypothesis, and perhaps the timing of when it arose or when the *Sp_81.2* C-terminal domain was presumably lost.

Some *C. darwini* specific motifs and cassettes that were highly abundant in MaSp spidroins (e.g., [GGP]3 or GGSGGGL) likely form helices that act to bind molecules together. This may contribute to increased ability to form composite fibers that may unite the mechanical properties of different spidroins and underlying the unusual biology of this species. For example, the web frame and radial threads are made of major ampullate silk that is impressively tough in *C. darwini* due to its higher extensibility, probably an adaptation to river-bridging webs [3]. The relatively high expression of flagelliform spidroins in major ampullate glands suggests that these spidroins fuse with major ampullate spidroins into a fiber that is both strong and unusually extendible—hence tough. Similarly, the unusually diverse aciniform spidroins might be connected to the incredible bridging capabilities of *C. darwini*, who uses a higher quantity of aciniform silk strands to form a sail-like terminus of the bridging silk, creating sufficient wind-drag to be carried over rivers and lakes [2].

Very recent (unpublished) work corroborates our genomic findings and provides even further proteomic details about spidroins found in manually drawn dragline silk and its toughness relative to other, related bark spiders found in Madagascar [50, 51]. Specifically, Kono et al. report that a set of non-spidroin products they referred to as SpiCE proteins contributes to the composite nature of the fibers and are important to tensile properties. Between both works, a picture is emerging that mixing multiple spidroins and other silk-associated proteins into a composite fiber can greatly impact and enhance mechanical properties and hence the function of silks. By integrating multiple genomic studies with ecological and biomechanical studies, a more complete picture of the silk production system of the Darwin's bark spider has been revealed.

## Supporting information

**S1 Fig. qPCR results for spidroins (females) for major ampullate, minor ampullate, and "other" glands.** Relative transcript abundance of spidroin targets plus controls across silk gland samples are presented, normalized to leg tissue samples and calculated using the 2-ΔΔCT method.
(PDF)

**S2 Fig. qPCR results for spidroins (females) for tubuliform, flagelliform, aggregate glands.** Relative transcript abundance of spidroin targets plus controls across silk gland samples are presented, normalized to leg tissue samples and calculated using the 2-ΔΔCT method.
(PDF)

**S3 Fig. qPCR results for spidroins (females) for epigynum, mature eggs, and ovaries + eggs.** Relative transcript abundance of spidroin targets plus controls across silk gland samples are presented, normalized to leg tissue samples and calculated using the 2-ΔΔCT method.
(PDF)

**S4 Fig. qPCR results for spidroins (females) for all cephalothorax, cephalothorax (without Chelicerae), and Chelicerae only.** Relative transcript abundance of spidroin targets plus controls across silk gland samples are presented, normalized to leg tissue samples and calculated using the 2-ΔΔCT method.
(PDF)

**S5 Fig. qPCR results for spidroins (females) for venom glands, pedipalps, and legs.** Relative transcript abundance of spidroin targets plus controls across silk gland samples are presented, normalized to leg tissue samples and calculated using the 2-ΔΔCT method.
(PDF)

**S6 Fig. qPCR results for spidroins (males) for abdomen, all silk glands, and cephalothorax.** Relative transcript abundance of spidroin targets plus controls across silk gland samples are presented, normalized to leg tissue samples and calculated using the 2-ΔΔCT method.
(PDF)

**S7 Fig. qPCR results for spidroins (males) for cephalothorax (w/o chelicerae), chelicerae and venom glands, and pedipalps.** Relative transcript abundance of spidroin targets plus controls across silk gland samples are presented, normalized to leg tissue samples and calculated using the 2-ΔΔCT method.
(PDF)

**S8 Fig. qPCR results for spidroins (males) for legs.** Relative transcript abundance of spidroin targets plus controls across silk gland samples are presented, normalized to leg tissue samples and calculated using the 2-ΔΔCT method.
(PDF)

**S9 Fig. Structure of the *C. darwini spinning apparatus*.** (a) Adult female spinnerets (ALS = anterior lateral spinneret, posterior median spinnerets = PMS, posterior lateral spinneret = PLS); (b) adult female ALS with marked piriform (pi) and major ampullate (map) spigots; (c) adult female PMS with marked tubuliform (tu), aciniform (ac) and minor ampullate (mip) spigots; (d) adult female PLS with marked tubuliform (tu) spigots; (e) adult female magnified PLS part with marked flagelliform (fl) and aggregate (ag) spigots; (f) adult male PLS with marked flagelliform (fl) and aggregate (ag) spigots.
(PDF)

**S1 Table. Samples used in the study.** Identifiers and accession details for samples used for experiments described in this study.
(XLSX)

**S2 Table. Sequencing libraries constructed for this study.** Details of barcodes, adaptors and samples used for constructing RNA, DNA, and SMRT Sequencing libraries.
(XLSX)

**S3 Table. Sequencing runs performed for this study.** Design details of sequencing performed for data collection.
(XLSX)

**S4 Table. DNA sequencing read files used in this study.** A list of the sequencing read file names and the type of library used in this study, including Illumina and PacBio technologies, with SRA accession numbers.
(XLSX)

**S5 Table. Genome assembly metrics.** Assembly metrics summarized for each type of assembly method as well as meta-assembly.
(XLSX)

**S6 Table. RNA sequencing read files used in this study.** A list of the sequencing runs and libraries used for transcriptome characterization, with SRA accession numbers.
(XLSX)

**S7 Table. Transcriptome assembly metrics.** Details of transcriptomic assembly measures (total transcripts, N50) as well as BUSCO summary for all isolates.
(XLSX)

**S8 Table. Input files used in the *C. darwini* genome annotation pipeline.** Details and sources of databases utilized for annotation mapping.
(XLSX)

**S9 Table. Genome annotation metrics.** A summary of annotation for various gene models of increasing stringency (Max, Standard, Silver, Gold, Platium).
(XLSX)

**S10 Table. Curated library of previously established spidroin genes used for analysis.** List of accessions for catalog of spidroins used for comparison and discovery of spidroins in *C. darwini*.
(XLSX)

**S11 Table. PCR primers used for amplicon generation towards SMRT sequencing.** List of primers used to generate amplicons for validating, gap closing, and merging putative spidroins characterized in the *C. darwini* genome.
(XLSX)

**S12 Table. Spidroin repeat motif type occurrences and coordinates.** Results for the locations and characterization of annotated motifs discovered from the spidroin motif painting analysis.
(XLSX)

**S13 Table. Spidroin repeat motif groups and descriptions for *Caerostris darwini*.** Summary of the motif groups discovered from the spidroin motif painting analysis.
(XLSX)

**S14 Table. Spidroin repeat motif types for *C. darwini* and sharing across genes and with *T. clavipes*.** Occurrence counts, motif lengths, and sharing details of motifs identified by the painting analysis.
(XLSX)

**S15 Table. Spidroin repeat type sharing across genes, classes, and species.** A summary of the frequency of sharing of motif groups across *C. darwini* and *T. clavipes*.
(XLSX)

**S16 Table. Spidroin repeat cassette type occurrences and coordinates.** Results for the locations and characterization of annotated cassettes discovered from the spidroin motif painting analysis.
(XLSX)

**S17 Table. Spidroin repeat cassette groups and descriptions for *Caerostris darwini* and *Trichonephila clavipes*.** Summary of the cassette groups discovered from the spidroin motif painting analysis.
(XLSX)

**S18 Table. Spidroin repeat cassette types for *Caerostris darwini* and *Trichonephila clavipes*.** Occurrence counts, motif lengths, and sharing details of cassettes identified by the painting analysis.
(XLSX)

**S19 Table. Spidroin repeat cassette type sharing across genes, classes, and species.** A summary of the frequency of sharing of cassette groups across *C. darwini* and *T. clavipes*.
(XLSX)

**S20 Table. List of qPCR primers used for silk-gland expression profiling.** Primer sequences used for the qPCR validation experiments performed for identified spidroins.
(XLSX)

**S1 Data. Amino acid sequences for the spidroins identified in *Caerostris darwini*.** This is a .fasta file that includes the set of sequences used for the motif painting analysis, plus the sequence for SpL_170.
(FSA)

## Acknowledgments

We thank our many colleagues who kindly shared their expertise, time, resources, and insights over the long history of this project, including J. Retief, and T. Orpin, J. Coddington, John Hogenesch, and members of the Voight Lab (past and present). We thank the MICET crew in Antananarivo, H. Rabarison, S. Rahanitriniaina, and O. Raberahona for their help in the field.

## Author Contributions

**Conceptualization:** Paul L. Babb, Linden Higgins, Ingi Agnarsson, Benjamin F. Voight.

**Data curation:** Paul L. Babb, Matjaž Gregorič, David N. Nicholson, Cheryl Y. Hayashi, Matjaž Kuntner, Benjamin F. Voight.

**Formal analysis:** Paul L. Babb, Nicholas F. Lahens, David N. Nicholson, Benjamin F. Voight.

**Funding acquisition:** Benjamin F. Voight.

**Investigation:** Paul L. Babb, Matjaž Gregorič, Cheryl Y. Hayashi, Matjaž Kuntner, Ingi Agnarsson, Benjamin F. Voight.

**Methodology:** Cheryl Y. Hayashi, Linden Higgins.

**Project administration:** Ingi Agnarsson, Benjamin F. Voight.

**Resources:** Matjaž Gregorič, Cheryl Y. Hayashi, Matjaž Kuntner, Ingi Agnarsson, Benjamin F. Voight.

**Software:** David N. Nicholson.

**Supervision:** Paul L. Babb, Benjamin F. Voight.

**Validation:** Paul L. Babb, Benjamin F. Voight.

**Visualization:** Paul L. Babb, Matjaž Gregorič, Matjaž Kuntner, Benjamin F. Voight.

**Writing – original draft:** Paul L. Babb, Benjamin F. Voight.

**Writing – review & editing:** Paul L. Babb, Matjaž Gregorič, Nicholas F. Lahens, David N. Nicholson, Cheryl Y. Hayashi, Linden Higgins, Matjaž Kuntner, Ingi Agnarsson, Benjamin F. Voight.

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
