## [Decision Letter · Decision Letter 0]

22 Feb 2022

PONE-D-22-01083Characterization of the genome and silk-gland transcriptomes of Darwin’s bark spider (Caerostris darwini)PLOS ONE

Dear Dr. Voight,

Thank you for submitting your manuscript to PLOS ONE. After careful consideration, we feel that it has merit but does not fully meet PLOS ONE’s publication criteria as it currently stands. Therefore, we invite you to submit a revised version of the manuscript that addresses the points raised during the review process. Minor changes of the manuscript are needed, as suggested by both reviewers.

We look forward to receiving your revised manuscript.

Kind regards,

Serena Aceto, Ph.D.

Academic Editor

PLOS ONE

Journal Requirements:

[We thank our many colleagues who kindly shared their expertise, time, resources, and insights over the long history of this project, including J. Retief, and T. Orpin, J. Coddington, John Hogenesch, and members of the Voight Lab (past and present). B.F.V. is grateful for support of the work from the Alfred P. Sloan Foundation (BR2012-087). We thank the MICET crew in Antananarivo, H. Rabarison, S. Rahanitriniaina, and O. Raberahona for their help in the field. Genomic assembly was conducted on the PMACS HPC infrastructure at the University of Pennsylvania, funded in part by NIH Special Instrumentation Grant 1S10OD012312-NIH.]

[Alfred P. Sloan Foundation (BR2012-087)

NIH Special Instrumentation Grant (1S10OD012312)]

Reviewers' comments:

Reviewer's Responses to Questions

**Comments to the Author**

1. Is the manuscript technically sound, and do the data support the conclusions?

Reviewer #1: Yes

Reviewer #2: Yes

2. Has the statistical analysis been performed appropriately and rigorously? 

Reviewer #1: Yes

Reviewer #2: Yes

3. Have the authors made all data underlying the findings in their manuscript fully available?

Reviewer #1: Yes

Reviewer #2: Yes

4. Is the manuscript presented in an intelligible fashion and written in standard English?

Reviewer #1: Yes

Reviewer #2: Yes

5. Review Comments to the Author

Reviewer #1: The manuscript “Characterization of the genome and silk-gland transcriptomes of Darwin’s bark spider (Caerostris darwini)” by P.L. Babb et al. is acceptable for publication in Plos One after minor revision.

The paper presents the whole sequencing of the genome of the Caerostris darwini spider with special attention to the spidroin genes. The same research group performed the first complete sequencing of the genome of a spider (Triconephila clavipes), which led to identification of the full set of spidroin genes in one species. Technically the work is irreproachable and the analysis highlights what the authors consider as the main findings of their work.

Consistently with this judgement, there are just a couple of suggestions with regard to the manuscript before being finally published.

1) Since the authors make an extensive comparison between the spidroin genes of Caerostris darwini and Triconephila clavipes, it would be helpful if they devote a few sentences to explain the phylogenetic relation between both species and their location in the Araenomorphae lineage.

2) The authors mention twice that “C. darwini...whose dragline is the toughest known biomaterial on Earth” (Abstract) and “…to create the toughest silks that have been documented to date [3]”. (Introduction). Undoubtedly, C. darwini spins one of the toughest spider silks, however, other spiders such Argiope aurantia (see Madurga et al., Scientific Reports 6:18991 (2016)) show comparable or even higher values of work to fracture. Therefore, I would recommend to downgrade the previous sentences to “… whose drageline is one of the toughest known biomaterials on Earth” and “…to create one of the toughest silks …”.

3) The authors rise the critical question of gene expression at several points along the manuscript: “These observations motivated further work …as well as their presumably complex expression across morphologically distinct silk glands” (Introduction) and “Previous work has demonstrated that spidroins assigned to a particular gene class are not exclusively transcribed in the silk gland in which the gene is assigned” (Results, Line 282). I fully agree with this view of the Authors and I think it is worth highlighting that similar conclusions are found through the proteomic analysis of protein expression of silk glands in orb-weaving spiders (Jorge et al., J. Exp. Zool. Part B, https://doi.org/10.1002/jez.b.23117).

Reviewer #2: Some minor suggestions:

1. As bulky amount of data was presented, descriptions linking to methods in the main text could be more specific, for example, “……for along with those we previously reported in T. clavipes (Methods)” (line 198), which part of the “Method”?

2. How to approach some of the data in detail? For example, “three C. darwini spidroins – Sp_81.2, SpL_133.2, and SpL_4399.2 – eluded assignment to any of the previously established classes based on sequence homology alone（line 187-188）” , but it’s not very direct to get these sequences and the relevant analysis, which is actually of real interest.

6. PLOS authors have the option to publish the peer review history of their article (what does this mean?). If published, this will include your full peer review and any attached files.

Reviewer #1: No

Reviewer #2: No

---

## [Author Response · Author response to Decision Letter 0]

14 Apr 2022

We thank the reviewers for a careful reading and comments on our submitted manuscript. Our responses to comments are provide below.

Reviewer #1: The manuscript “Characterization of the genome and silk-gland transcriptomes of Darwin’s bark spider (Caerostris darwini)” by P.L. Babb et al. is acceptable for publication in Plos One after minor revision.

The paper presents the whole sequencing of the genome of the Caerostris darwini spider with special attention to the spidroin genes. The same research group performed the first complete sequencing of the genome of a spider (Triconephila clavipes), which led to identification of the full set of spidroin genes in one species. Technically the work is irreproachable and the analysis highlights what the authors consider as the main findings of their work.

Author Response: We very much thank the reviewer for their support for our work.

Consistently with this judgement, there are just a couple of suggestions with regard to the manuscript before being finally published.

1) Since the authors make an extensive comparison between the spidroin genes of Caerostris darwini and Triconephila clavipes, it would be helpful if they devote a few sentences to explain the phylogenetic relation between both species and their location in the Araenomorphae lineage.

Author Response: We agree, and have added the following text to the discussion:

Recent work has greatly expanded the number of spider genomes available for analysis, and thus considering how C. darwini relates to T. clavipes bears some brief discussion. Currently, C. darwini and T. clavipes are catalogued as representatives of the same family, Araneidae [25]. However, Kuntner et al. [26] consider Trichonephila as member of Nephilidae, and Caerostris as member of Araneidae, both families being true orb-weavers, Orbipurae. This is because Araneidae as currently circumscribed is poorly defined and ancient [26]. Regardless of which family classification is being followed, the current evidence dates the most recent common ancestor of Caerostris darwini and Trichonephila clavipes to between 62 and >200 million years ago, albeit with large confidence intervals [18,19,26]. 

2) The authors mention twice that “C. darwini...whose dragline is the toughest known biomaterial on Earth” (Abstract) and “…to create the toughest silks that have been documented to date [3]”. (Introduction). Undoubtedly, C. darwini spins one of the toughest spider silks, however, other spiders such Argiope aurantia (see Madurga et al., Scientific Reports 6:18991 (2016)) show comparable or even higher values of work to fracture. Therefore, I would recommend to downgrade the previous sentences to “… whose drageline is one of the toughest known biomaterials on Earth” and “…to create one of the toughest silks …”.

Author Response: We agree, and have edited the abstract and introductory text as suggested.

3) The authors rise the critical question of gene expression at several points along the manuscript: “These observations motivated further work …as well as their presumably complex expression across morphologically distinct silk glands” (Introduction) and “Previous work has demonstrated that spidroins assigned to a particular gene class are not exclusively transcribed in the silk gland in which the gene is assigned” (Results, Line 282). I fully agree with this view of the Authors and I think it is worth highlighting that similar conclusions are found through the proteomic analysis of protein expression of silk glands in orb-weaving spiders (Jorge et al., J. Exp. Zool. Part B, https://doi.org/10.1002/jez.b.23117).

Author Response: Thanks to the reviewer for this very recent reference! We have included this reference and a note of this point in the discussion. 

In results: “… with proteomic analyses also indicating diverse protein expression across silk glands [24]”

In discussion: “… On this particular point, we note very recent work which reached similar conclusions from analysis of protein expression of silk glands in orb-weaving spiders as well [24], further supporting that diverse spidroin expression is functionally consequential.”

Reviewer #2: Some minor suggestions:

1. As bulky amount of data was presented, descriptions linking to methods in the main text could be more specific, for example, “……for along with those we previously reported in T. clavipes (Methods)” (line 198), which part of the “Method”?

Author Response: Where possible we have indicated a sub-header where in methods details could be found.

2. How to approach some of the data in detail? For example, “three C. darwini spidroins – Sp_81.2, SpL_133.2, and SpL_4399.2 – eluded assignment to any of the previously established classes based on sequence homology alone (line 187-188)" but it’s not very direct to get these sequences and the relevant analysis, which is actually of real interest.

Author Response: We have now provided the amino acid sequences of spidroins we initially characterized in C. darwini (Supplementary Data 1). The homology characterization experiments are results of interpreting BLAST results using these sequences against the catalog of spidroins that we previously curated (described in methods with details of sequences available in Supplementary Table 10). Spidroins were defined and identified by (i) the presence of coding repetitive motifs, (ii) blast protein / nucleotide sequence alignments of N and C-terminal domains with the N/C terminal domains of established, canonical spidroins, (iii) expression in silk glands, and (iv) manual review and guidance from domain-expert collaborators. We have added details to methods (L210 – 223).

---

## [Decision Letter · Decision Letter 1]

4 May 2022

Characterization of the genome and silk-gland transcriptomes of Darwin’s bark spider (Caerostris darwini)

PONE-D-22-01083R1

Dear Dr. Voight,

We’re pleased to inform you that your manuscript has been judged scientifically suitable for publication and will be formally accepted for publication once it meets all outstanding technical requirements.

Kind regards,

Serena Aceto, Ph.D.

Academic Editor

PLOS ONE

Additional Editor Comments (optional):

Reviewers' comments:

Reviewer's Responses to Questions

**Comments to the Author**

1. If the authors have adequately addressed your comments raised in a previous round of review and you feel that this manuscript is now acceptable for publication, you may indicate that here to bypass the “Comments to the Author” section, enter your conflict of interest statement in the “Confidential to Editor” section, and submit your "Accept" recommendation.

Reviewer #1: All comments have been addressed

Reviewer #2: All comments have been addressed

2. Is the manuscript technically sound, and do the data support the conclusions?

Reviewer #1: Yes

Reviewer #2: Yes

3. Has the statistical analysis been performed appropriately and rigorously? 

Reviewer #1: Yes

Reviewer #2: Yes

4. Have the authors made all data underlying the findings in their manuscript fully available?

Reviewer #1: Yes

Reviewer #2: Yes

5. Is the manuscript presented in an intelligible fashion and written in standard English?

Reviewer #1: Yes

Reviewer #2: Yes

6. Review Comments to the Author

Reviewer #1: The paper is acceptable for publication in its present form. The authors have addressed all my previous concerns.

Reviewer #2: The authors presented genomic dissection of Darwin’s bark spider, Caerostris darwini, and focused on the new putative spidroin genes. The data has been clearly and systematically presented.

This manuscript is therefore suggested for direct acceptance.

7. PLOS authors have the option to publish the peer review history of their article (what does this mean?). If published, this will include your full peer review and any attached files.

Reviewer #1: No

Reviewer #2: No

---

## [Editor Report · Acceptance letter]

26 May 2022

PONE-D-22-01083R1 

Characterization of the genome and silk-gland transcriptomes of Darwin’s bark spider (*Caerostris darwini*) 

Dear Dr. Voight:

I'm pleased to inform you that your manuscript has been deemed suitable for publication in PLOS ONE. Congratulations! Your manuscript is now with our production department. 

Kind regards, 

on behalf of

Dr Serena Aceto 

Academic Editor

PLOS ONE